# FREE-VIEW ROBOT MANIPULATION: VISUOMOTOR POLICY BY CALIBRATION DIFFUSION

## ABSTRACT

Visuomotor policies have demonstrated great potential in robot manipulation tasks. However, current robot manipulation tasks are often observed from fixed viewpoints. Once the viewpoints change, the trained policy becomes ineffective. This limitation curbs the generalization of robot manipulation and impedes its application. To address this issue, we make a comprehensive study by presenting novel free-view manipulation tasks that enables the robot to perform actions from any viewpoint. Firstly, we construct a free-view dataset, which encompasses 10 tasks with over 5,000 episodes sourced from the Isaac Sim simulation and real-world. Each episode records robot manipulation behaviors from different viewpoints. Secondly, we propose a calibration diffusion policy, which utilizes an additional calibration network to enhance the adaptability of the diffusion policy to different viewpoints. In particular, we adopt two-stage curriculum training to make the calibration diffusion policy converge rapidly. Finally, we conduct a wealth of experiments on the free-view dataset. The obtained results demonstrate the effectiveness of the calibration diffusion policy. This also means that we have built a new benchmark for free-view manipulation.

## 1 INTRODUCTION

Visuomotor policy action generation for robots in unstructured environments has become a major research focus in embodied intelligence Brohan et al. (2022; 2023); Zhao et al. (2023); Chen et al. (2024b). When processing specific tasks, the robot, considering its current state, captures environment information via the camera, generates the next action policy, and conducts manipulation skills in the physical world. However, current robot action generation policies are strongly related to the fixed robot's observation viewpoint as illustrated in Figure 1(a). Once there is an offset in the camera installation or the carefully calibrated camera is moved (see Figure 1(c)),

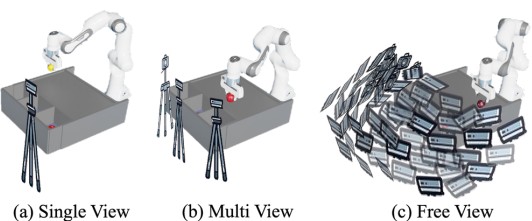

(a) Single View     (b) Multi View     (c) Free View

Figure 1: Diverse Categories of Camera Viewpoints. (a) Single View: only one camera is fixed at one position to observe motion behavior. (b) Multi View: multiple cameras observe motion behavior at different fixed positions. (c) Free View: cameras observe motion behavior from any viewpoint.

the trained manipulation policies will no longer be applicable Yuan et al. (2024); Ze et al. (2024a). This problem seriously affects the implementation of robot experiments and the deployment of well-trained manipulation skills among different robots.

Among various visuomotor policies, the diffusion policy Chi et al. (2023) is particularly remarkable. It can exhibit robust generalization capabilities for individual specific tasks using merely a limited number of interactive data samples. Currently, it has been widely applied in various manipulation tasks Chi et al. (2023); Walke et al. (2023); Ren et al. (2024); Dasari et al. (2024) including dual-arm manipulation Mu et al. (2024); Drolet et al. (2024) and human-robot collaboration Ng et al. (2023); He et al. (2024). However, when using these well-trained policies to carry out tasks, it is often necessary for the camera's viewpoint to be exactly the same as that in the dataset. Otherwise, the effectiveness would decline significantly Pang et al. (2025). Even for the same task, changes in the

camera's viewpoint are very likely to disrupt the trained policy. The fundamental reason is that the calibration relationship between the camera and the robot has changed, but the visuomotor policy is unable to adapt to this change.

To tackle this problem, we introduce novel free-view robot manipulation tasks in this work. As shown in Figure 1(a) and (b), the existing robot behaviors are typically observed from either a single or multiple fixed viewpoints. This limitation causes the policies to be only able to learn the robot manipulation behaviors from a limited number of restricted viewpoints and to lack generalization ability. In a free-view manipulation task, the camera observes from a random viewpoint each time. The visuomotor policy can break free from the constraints of the viewpoints and successfully complete manipulation tasks. Moreover, we establish a new free-view manipulation dataset with multiple baselines. We employ the advanced Isaac Sim NVIDIA (2021) simulation environment to construct a comprehensive dataset. Utilizing conventional robot motion planning methods, we accomplish eight different simulation tasks, which include multiple actions such as pushing, picking, pulling, sorting, and striking. Each episode is observed from different viewpoints and records images, motion trajectory and calibration relationships. In our Free View dataset, we collected over 5000 episodes, each with a unique viewpoint.

To endow robots with free-view manipulation capabilities, we reconsider the camera calibration relationship, and have an idea that a rough camera calibration can be used as an extra condition. Inspired by ControlNet Zhang et al. (2023), we propose a novel Calibration Diffusion Policy with condition control that can generate robot manipulation actions from different observation viewpoints. Specifically, in the first part, the main backbone still relies on the diffusion policy Chi et al. (2023). By encoding the image and depth from different viewpoints and the current state of the robot, the diffusion model is utilized to generate the next action. In the second part, we build a twin calibration network to extract features of the camera calibration from different camera coordinates to the robot coordinate system. During the process of supervised fine-tuning, the camera calibration features are integrated into the backbone. This integration allows the diffusion model to discern the observation viewpoint, thereby facilitating the generation of more precise actions. In addition, we also deploy the two-stage training method of ControlNet. In the first stage, we only train the basic diffusion policy, and in the second stage, we train the calibration network to extract calibration features. Based on the two-stage curriculum training, we only need a small amount of additional training data, so that the policy can be better adapted to the variations of viewpoints.

The main contributions of this paper are summarized as follows:

**(1)** We present novel free-view manipulation tasks and build a free-view dataset both in the Isaac Sim simulation environment. For each manipulation data, the camera viewpoints are all different, enabling the visuomotor policies to learn free-view generalization capabilities.

**(2)** To enhance the robot manipulation capabilities in free-view, we propose a Calibration Diffusion Policy method, which is trained rapidly via curriculum learning. It also employs the camera calibration relationship to adjust actions for viewpoint changes.

**(3)** Numerous and rich experiments on the free-view dataset demonstrate that compared with previous methods, the Calibration Diffusion Policy exhibits stronger generalization ability and higher reliability. This also means that we have built a new benchmark for free-view manipulation.

## 2 RELATED WORKS

### 2.1 ROBOT MANIPULATION

Robot manipulation research has achieved significant advances due to the emergence of artificial intelligence and embodied intelligence concepts. Traditional sampling-based methods LaValle (1998); Karaman & Frazzoli (2011) often yield lengthy trajectories. Optimization-based methods Huang et al. (2024); Jin et al. (2024) require clearly defining the task objectives in advance. Recently, learning-based methods, such as reinforcement learning Eysenbach et al. (2022); Yuan et al. (2024); Chen & Rojas (2024); Liang et al. (2024), imitation learning Xie et al. (2024); Chen et al. (2024a); Zhao et al. (2023); Shafiullah et al. (2022) and generative models Chi et al. (2023); Ze et al. (2024b); Wen et al. (2024), are gradually being applied to robot manipulation tasks. These methods can be trained with a small amount of data and then directly generate manipulation trajectories in the similar scenarios, providing a more efficient and adaptable solution for practical applications. In this paper,

we focus on the generative-model-based method with few-shot data. By means of the calibration diffusion policy, this method significantly enhances the generalization ability of robots to complete manipulation tasks under different viewpoints.

For generative-model-based methods, diffusion policy Chi et al. (2023) has demonstrated remarkable generalization ability, and thus has been extremely widely applied in various complex and diverse robot manipulation tasks Mishra et al. (2023); Walke et al. (2023); Zhang et al. (2024). Based on image and depth data, Ma et al. Ma et al. (2024) proposed a hierarchical diffusion strategy and successfully applied the diffusion policy to the field of multi-task manipulation. Sridhar et al. Sridhar et al. (2024) skillfully applied the diffusion policy to the exploration tasks of mobile robots, providing new methods for robot exploration. Furthermore, as flow matching Lipman et al. (2023); Liu et al. (2023) gains broader adoption in generative models, several studies have extended its application to robotic manipulation tasks and proposed the concept of flow policy Zhang et al. (2025); Fang et al. (2025). Flow Policy is based on ordinary differential equations (ODEs) and offers a significant advantage in inference speed over Diffusion Policy. 3D Diffusion Policy Ze et al. (2024b) (DP3) leverages point cloud data to generate robust manipulation actions. However, these point cloud-based works Xue et al. (2025); Ze et al. (2024a) need extra pre-processing like point cloud transformation, segmentation, and removal of redundant point clouds. In contrast, this work uses only raw image and depth data (easier to access), and then presents a calibration net for free-view manipulation tasks.

## 2.2 CAMERA VIEWPOINTS

We categorize the existing visuomotor policies according to the calibration relationship between cameras and robots. A minority employs the "eye in hand" camera installation method Zhang et al. (2024); Yang et al. (2025); Yao et al. (2025), where the camera is fixed to the End Effector of the robot arm, ensuring that it moves in tandem with the End-Effector. In this configuration, the calibration relationship between the camera and the End Effector remains constant. Once the end-effector is replaced or the installation position of the camera is adjusted, it is necessary to re-collect data for training. Conversely, the majority of policies opt for the "eye to hand" camera installation method Walke et al. (2023); Ma et al. (2024); ?, where the camera's viewpoint is independent of the robot, and it does not move with the robot. In this configuration, the calibration relationship between the camera and the robot base is fixed. However, in real-world research applications, it is rather difficult to ensure that the observation viewpoints of cameras remain consistent all the time. The existing policies are unable to accomplish the same manipulation task of the robot under different camera viewpoints. Some works Luo et al. (2024; 2023); Bharadhwaj et al. (2024) mix the "eye in hand" camera and "eye to hand" cameras to achieve better manipulation capabilities. Nevertheless, these fixed mixed camera viewpoints are still unable to mitigate the instability arising from camera viewpoint changes.

## 3 DATASET

In this paper, we present an innovative free-view robot manipulation dataset in both simulation and real-world environments, which includes ten scene tasks. For simulation scenes, we utilize Isaac Sim NVIDIA (2021) and cameras can be directly generated at free positions. The calibration relationship between the cameras and the robot can be directly obtained during the simulation. We use traditional kinematics and planning methods to enable the robot to complete manipulation tasks, and observe the behavior with freely positioned cameras. For real-world scenes, we employ the Franka with programmed teaching to perform tasks. We use a camera group composed of three RealSense D455 cameras. The motion trajectory of the robot in each instance is captured by the three cameras. Prior to each execution of a taught trajectory, camera positions are freely repositioned, and the relationship between camera and robot is calibrated via a checkerboard. In this case, each episode in the dataset features a distinct observation viewpoint and an exclusive calibration relationship.

Based on Figure 2, we specifically introduce 10 types of tasks in our free-view dataset. Each detailed task introduction can be found in Appendix. A.1. Compared with other robot manipulation datasets in Table 1, we adopt the "eye to hand" camera installation method, which better conforms to human observation habits. In each episode, we register the exclusive calibration relationship, and record robot states, the images and depth in sequence. Different from the previous single-view or multi-view collections at fixed positions, the visual data in each episode of the free-view dataset is acquired from any view within the observation space. The tasks we designed not only take into account simple

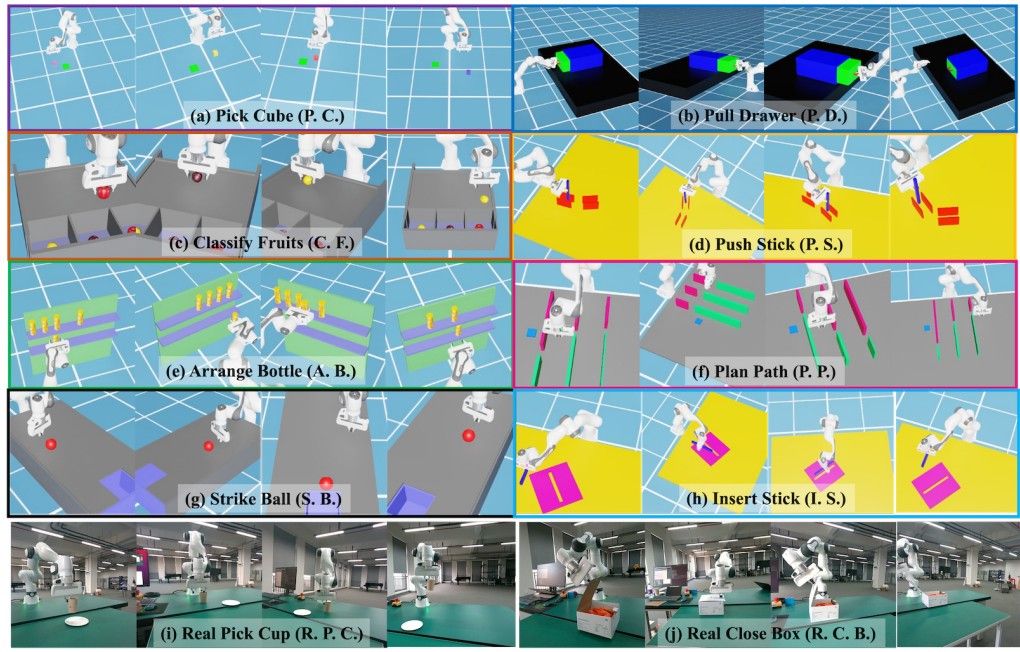

Figure 2: Visualization of our free-view dataset. Each episode in the dataset contains serialized images, depths, end effector positions, and calibration parameters. Here, we present four images from four episodes for each task. It can be observed that the camera viewpoints differ across different episodes.

Table 1: Comparison to existing various datasets for robot manipulation. The symbol * represents skills, which means that there are differences between tasks and skills in the dataset. When the same skill interacts with diverse objects, it can be considered as different tasks. The symbol $+$ represents extra adjustable cameras, which means that there are extra adjustable cameras that observe the robot's manipulation from different viewpoints.

| Dataset | Episodes | Tasks | Camera | Calibration | View |
|---|---|---|---|---|---|
| Robomimic Mandlekar et al. (2021) | 2,200 | 8 | Eye to Hand | No | Single |
| RT-1 Brohan et al. (2022) | 130,000 | 700+ | Eye to Hand | No | Single |
| Berkeley Autolab Chen et al. | 896 | 4 | Mixed | No | Multi |
| Taco Play Rosete-Beas et al. (2022) | 3,242 | 6 | Mixed | No | Multi |
| Berkeley Bridge Walke et al. (2023) | 60,096 | 13* | Mixed | No | Multi$+$ |
| NYU Franka Play Cui et al. (2022) | 456 | 6 | Eye to Hand | No | Multi |
| Stanford HYDRA Belkhale et al. (2023) | 550 | 4 | Mixed | No | Multi |
| Cable Routing Luo et al. (2024) | 1,442 | 1 | Mixed | No | Multi |
| MobileALOHA Fu et al. (2024) | 276 | 7 | Mixed | No | Multi |
| FMB Luo et al. (2023) | 22,550 | 6* | Mixed | Yes | Multi |
| Ours | 6,188 | 10 | Eye to Hand | Yes | **Free** |

picking and placing, but also involve obstacle avoidance planning, tool operation, decision, and prediction. These call for policies with a profound understanding of the entire scene. As far as we know, this may be the first free-view dataset with exclusive calibration parameters for each episode.

## 4 METHOD

Our calibration diffusion policy employs an additional calibration network architecture to enhance the effectiveness of the diffusion policy Chi et al. (2023) in the free-view. First, we introduce a comprehensive calibration diffusion framework, along with the corresponding input and output data in Sec. 4.1. Second, we introduce the calibration network details and how we apply the calibration net to the basic diffusion policy in Sec. 4.2. Third, we elaborate on our training process in Sec. 4.3.

### 4.1 FRAMEWORK

Our Calibration Diffusion framework is developed based on the diffusion model Ho et al. (2020) and the diffusion policy Chi et al. (2023). As shown in Figure 3, it consists of a backbone network, a calibration network, and two conditional branch networks for handling historical information.

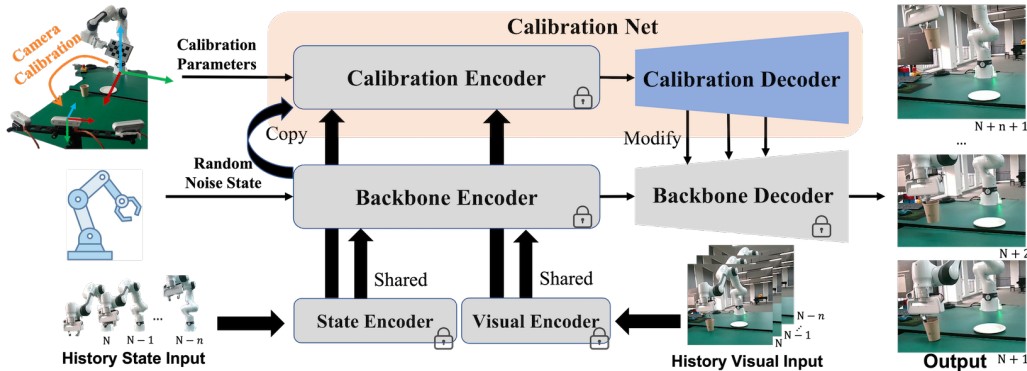

Figure 3: Overview of our Calibration Diffusion Policy. It mainly contains two twin networks. One is the backbone network that receives random noise states, and the other is the calibration network that receives calibration parameters. Additionally, two conditional branch networks receive historical robot states and visual information as shared conditions and then feed them into the main twin networks. With the help of the calibration network, the backbone network removes state noise and generates future robot trajectory states.

Mathematically, it is formulated as:

$$
\begin{aligned}
S_{T+} &= F(S^{noise}, C | S_{T-}, I_{T-}), \\
&= \mathbf{U}(S^{noise} | \mathbf{M}(S_{T-}), \mathbf{R}(I_{T-})) \oplus \mathbf{U}'(C | \mathbf{M}(S_{T-}), \mathbf{R}(I_{T-})),
\end{aligned}
\tag{1}
$$

where:

- $S_{T+}$ represents the future trajectory state predicted by the policy.
- $S^{noise}$ is the initial random gaussian noise in the diffusion process.
- $C$ is the camera calibration parameter from the current viewpoint.
- $S_{T-}$ and $I_{T-}$ are the historical trajectory state of the robot and the image observed by the camera, which are conditions for the diffusion policy Chi et al. (2023).

In detail,

- $\mathbf{U}$ denotes the backbone network with downsampling as the encoder and upsampling as the decoder, which is consistent with diffusion policy design.
- $R$ denotes a ResNet, used to extract consistent visual features from historical visual information.
- $M$ denotes a multi-layer perceptron (MLP), used to extract features from historical trajectory states.
- $\mathbf{U}'$ represents the calibration network we proposed. Its encoder weights are fully shared with the backbone network's encoder, and its decoder has the same structure as the backbone network's decoder but requires retraining. The conditional information is integrated during the encoding processes of both the backbone network and the calibration network, and it is shared between these two components.
- $\oplus$ denotes that the hierarchical output of the calibration network's decoder is fed back to the backbone network's decoder to modify the final output.

**Forward Diffusion Process**: The forward diffusion process is divided into $T$ time steps. At any time step $t$, random Gaussian noise is gradually added to the label-trajectory state $S_{N+}^{t-1}$. The calibration diffusion model predicts the trajectory state at the final time $T$ to make it closely follow a Gaussian distribution. The diffusion process $f$ can be represented by a formula as:

$$
S_{N+}^{t+1} = f(S_{N+}^{t}, C | S_{N-}, I_{N-}), \text{ to make } S_{N+}^{T} \sim \mathcal{N}(0, 1),
\tag{2}
$$

where the superscript $t$ represents the time step of the diffusion process.

**Denoising Sampling Process**: In the denoising sampling process, for a random Gaussian noise trajectory state, the calibration diffusion model gradually removes the noise to reverse and generate a new trajectory state. For any time $t$ among the $T$ time steps, the denoising process $f^{-1}$ can be represented by a formula as:

$$
S_{N+}^{t-1} = f^{-1}(S_{N+}^{t}, C | S_{N-}, I_{N-}).
\tag{3}
$$

$f^{-1}$ can be regarded as the inverse process of $f$. When $t = T$, $S_{N+}^{T}$ is a random noise trajectory. When $t = 0$, $S_{N+}^{0}$ is the trajectory state at the next moment predicted after the denoising process.

## 4.2 CALIBRATION NET

We have introduced a calibration network, which is used to extract the features of calibration parameters. Inspired by ControlNet Zhang et al. (2023) in the diffusion model, it can be regarded as the siamese network of the basic diffusion policy. But the input of the calibration network is the calibration parameters from the robot coordinate system to the camera coordinate system, Calibration parameters vary according to different viewpoints. Specifically, when the observation viewpoints are close to each other, the values of the corresponding calibration parameters are also similar. The initial weights of the calibration network are completely copied from the backbone network of the trained diffusion policy. Visual and state features are input as shared condition information into the calibration network. During the calibration network training, the weights of the encoder are frozen and will not be updated. Instead, only the decoder of the calibration network updates its weights.

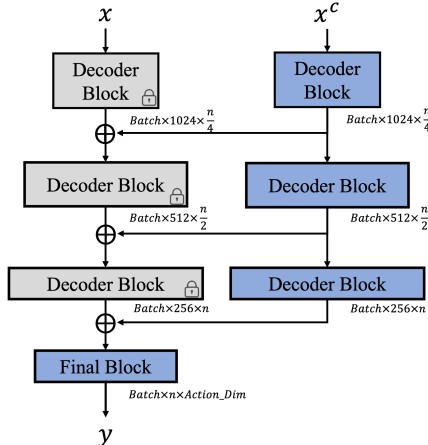

Figure 4: Decoder modification structure within calibration diffusion policy, which contains three levels hierarchical fusion.

The decoder of the calibration network will modify the output of the backbone through structured feedback. This allows the policy to better adapt to observations from different viewpoints, resulting in more accurate actions. As shown in Figure 4, $x$ is the encoding feature of the basic diffusion policy, and $x^c$ is the encoding feature of the calibration network. The decoder contains three level decoder blocks that reduce feature channels but increase dimensionality. Each decoder block consists of 2 ResNet modules. Each output of the decoder block of the calibration network is added to the corresponding output of the decoder block of the backbone network. The final block will readjust the weights during the calibration network training process to receive the outputs of the two parts. These hierarchical fusion structure enables the decoder to better utilize these shallow and deep features. Meanwhile, during the training process, hierarchical fusion allows the gradients from the decoder to be directly transmitted to the shallow and middle layers of the encoder, avoiding the "gradient vanishing" problem and enhancing training stability.

## 4.3 TRAINING DETAILS

In our free-view dataset, the state $S$ of the robot includes the end-effector positions $[p_x, p_y, p_z]$ and quaternions $[q_w, q_x, q_y, q_z]$, as well as the positions of the two-finger gripper $[g_l, g_r]$. The camera calibration parameter $C$ is the transformation relationship from the robot coordinate system to the camera coordinate system, which can also be represented by positions $[p_x^c, p_y^c, p_z^c]$ and quaternions $[q_w^c, q_x^c, q_y^c, q_z^c]$. We add zeros after the parameters of $C$ to make the dimensions of $S$ and $C$ consistent. The image size is $256 \times 256$, and the depth map is of the same size. We set the trajectory length $n = 8$, which means that observations from eight historical sequences are used to generate actions in the next eight future sequences.

We conduct the model training on RTX 4090 GPUs, and adopt a two-stage training approach. In the first stage, we solely train the basic diffusion policy including the backbone network based on samples from different viewpoints over 1,200 epochs. In the second stage, we copy the trained weights of the backbone to the calibration network. Then, we freeze most of the network weights and only update the weights of the decoder in the calibration network over an additional 200 epochs. In most cases, we employ epsilon prediction and DDPM noise scheduler for better generalization. The timestep number in the diffusion process is set to 100. In addition, the batch size and the learning rate are set to 128 and 5e-4, respectively.

## 5 EXPERIMENTS

### 5.1 EXPERIMENT SETTING

On various tasks of the free-view dataset, we reproduce some existing visuomotor policies to build baselines and verify the performance of our calibration diffusion policy method. Most policies utilize

RGBD data. We endeavor to keep the BC-T Florence et al. (2022), ACT Zhao et al. (2023), and Diffusion Policy (DP) Chi et al. (2023) methods as consistent as possible with the source code. The multi-view input of ACT has been modified to a single free-view input. We also refer to the 2D version of the Flow Policy (FP) Fang et al. (2025) method and reconstruct it to adapt to the free-view dataset. The observation sequences of BC-T, DP and FP are set to 8, just the same as our calibration diffusion policy. All input images are set to the resolution of 256×256. We also replicate the DP3 Ze et al. (2024b) that uses point cloud data. Specifically, we employ additional transformations to convert the RGBD data into point clouds. However, for fairness, we do not irrelevant remove the background through point cloud segmentation.

For evaluation, we define a free-view and free-object evaluation situation. On one hand, the observation viewpoint is freely generated within the observation space. On the other hand, the object to be manipulated is also randomly generated within the operation range. To ensure fairness, in most experiments of this paper, the policies are trained with 300 episodes, and each policy is tested for extra 50 episodes with the same random seed on each task.

Table 2: Free-view free-object results of different tasks (300 episodes).

| Method | P. C. | P. D. | C. F. | P. S. | A. B. | P. P. | S. B. | I. S. | R. P. C. | R. C. B. |
|---|---|---|---|---|---|---|---|---|---|---|
| BC-T | 0% | 0% | 18% | 26% | 6% | 0% | 6% | 2% | / | / |
| ACT | 14% | 4% | 8% | 10% | 0% | 4% | 2% | 0% | / | / |
| DP | 46% | 54% | 32% | 60% | 52% | 14% | 20% | 16% | 38% | 52% |
| FP | 36% | 44% | 40% | 54% | 54% | 12% | 24% | 8% | 38% | 44% |
| DP3 | 54% | 22% | **56%** | 66% | 42% | 2% | **40%** | 0% | / | / |
| **Cali. DP** | **64%** | **58%** | 52% | **80%** | **74%** | **42%** | 28% | **22%** | **46%** | **58%** |

## 5.2 MAIN RESULTS

The detailed comparative results are shown in Table 2. It can be seen that ACT does not seem to achieve the expected results. We believe that although ACT relies on data from multi-view, the method itself does not possess the same level of generalization ability as generative models, thus failing to adapt to free-view tasks. DP has been considered to have strong generalization capabilities. FP demonstrates results similar to those of DP. This is due to the fact that they share the same network structure; the distinction comes down to their denoising methods. Our Calibration DP improves the performance of free-view tasks by using an additional calibration network based on the basic DP. But for some high-precision and highly challenging tasks **I. S.**, it is still difficult to achieve the desired results. DP3 uses point clouds and performs quite well in the **C.F.** task and the **S.B.** task. Essentially, once the point cloud is transformed into the robot coordinate system, it has no direct connection with the viewpoint. However, different viewpoints result in varying degrees of point cloud data sparsity, and this inconsistency in point cloud data may be the reason why DP3 performs worse than Calibration DP.

In Figure 5, we provide visualization examples in the situation of free-view and free objects, which are all generated by our calibration diffusion policy. In different tasks, the calibration DP can withstand the impact brought about by viewpoint changes and complete manipulation. It is worth noting that in the **P.D.** task and the **S.B.** task, there is a significant change in viewpoints. The calibration diffusion policy can still adapt to these variations of viewpoints. More visual comparisons of DP and our Calibration DP can be found in the Appendix A.2, which show that Calibration DP exhibits greater robustness against viewpoint variations.

**Data Ablation.** We also conduct a series of comprehensive ablation experiments to demonstrate the effectiveness of the Calibration Net. In Figure 6, we analyze the role that training dataset size plays in **P. D.** and **P. S.** two specific tasks. As the data size grows, both the basic DP and our calibration DP see a significant boost in task success rates. Although increasing the number of training epochs for the basic DP can lead to a slight increase, our calibration DP can achieve a much more significant improvement with fewer additional epochs. More experiments A.3 also verify this trend. Further analysis reveals that the advantages of calibration DP become more prominent when dealing with limited data. With the training data growing in size, the success rate of the basic DP gradually converges towards that of the calibration DP. We believe that although the calibration DP has a stronger advantage in small data, the data has the greatest impact on the visuomotor policy in the free-view, and more data can make the policy have a stronger generalization ability.

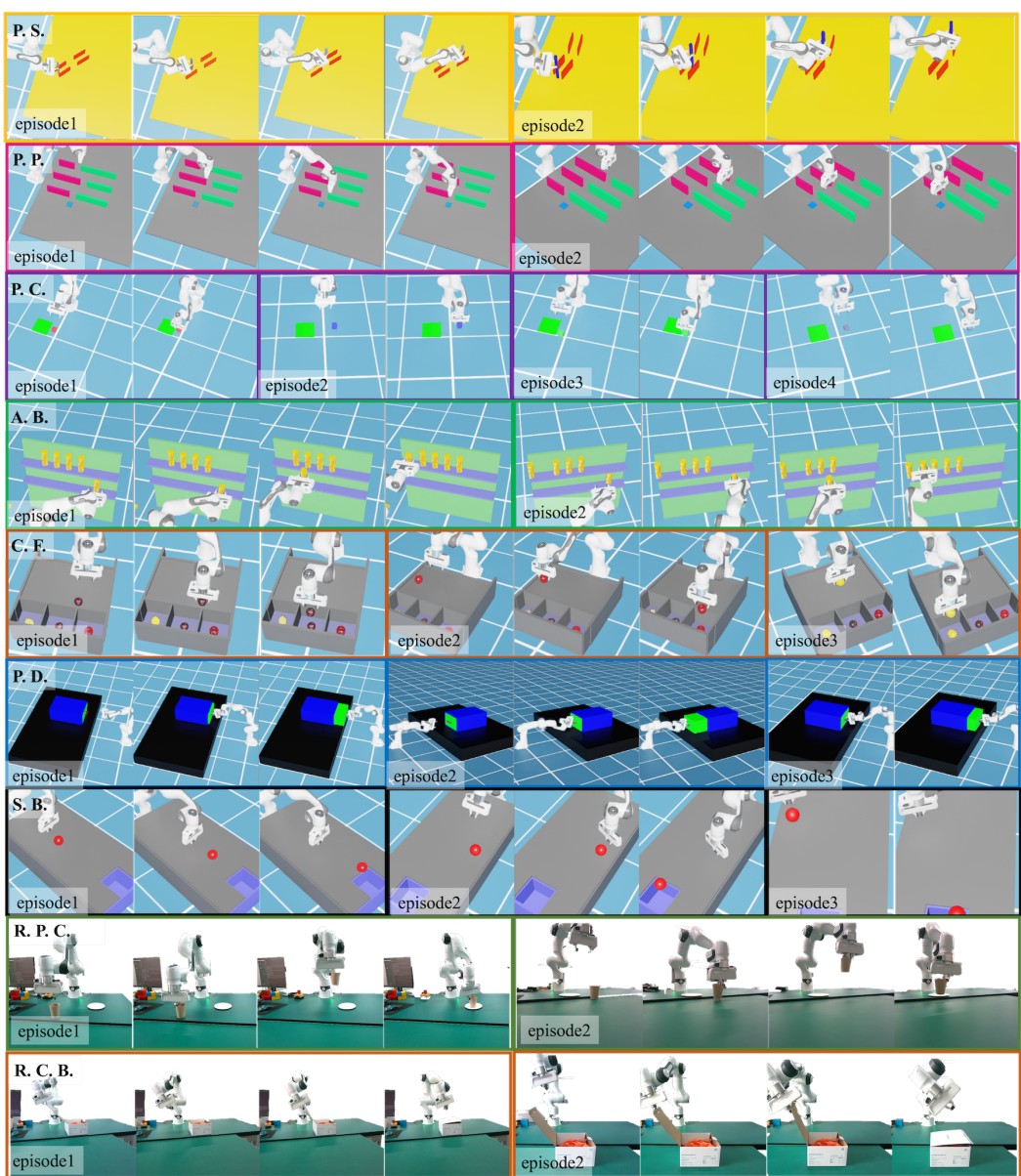

Figure 5: Free-view visualization results generated by using our Calibration Diffusion Policy. For each task, the viewpoints differ among individual episodes, and our proposed Calibration Diffusion Policy method can adapt to these viewpoint differences to complete robotic manipulation tasks.

Table 3: Ablation results for calibration net.

| Method | P. D. | P. S. | A. B. |
|---|---|---|---|
| Cali. DP from scratch | 30% | 58% | 52% |
| Cali. DP without freezing | 44% | 70% | 60% |
| Cali. DP with rand input | 48% | 74% | 68% |
| Cali. DP | **58%** | **80%** | **74%** |

Table 4: Ablation results for observation space.

| Obs. Space | | $x \in [1, 1.8]$ and $z \in [1.2, 1.8]$ | | |
|---|---|---|---|---|
| | | $|y| \le 0.5$ | $|y| \le 1$ | $|y| \le 2$ |
| P. C. | DP | 70% | 46% | 24% |
| | Cali. DP | **86%** | **64%** | **32%** |
| C. F. | DP | 48% | 32% | 20% |
| | Cali. DP | **60%** | **52%** | **26%** |

**Structure Ablation.** Then we explore the ablation experiments conducted on the calibration DP structure in Table 3. We attempt to train the Calibration DP directly from scratch without using the two-stage curriculum training. The results indicate that this approach yields notably lower training efficiency and inferior performance compared to the basic DP. We also make attempts to conduct the training without freezing the weights of the basic DP. Additionally, we use random inputs to replace the calibration parameters as the input for the Calibration Net. It appears that the calibration network itself indeed brings about incremental improvements to the results, indicating its significant role in

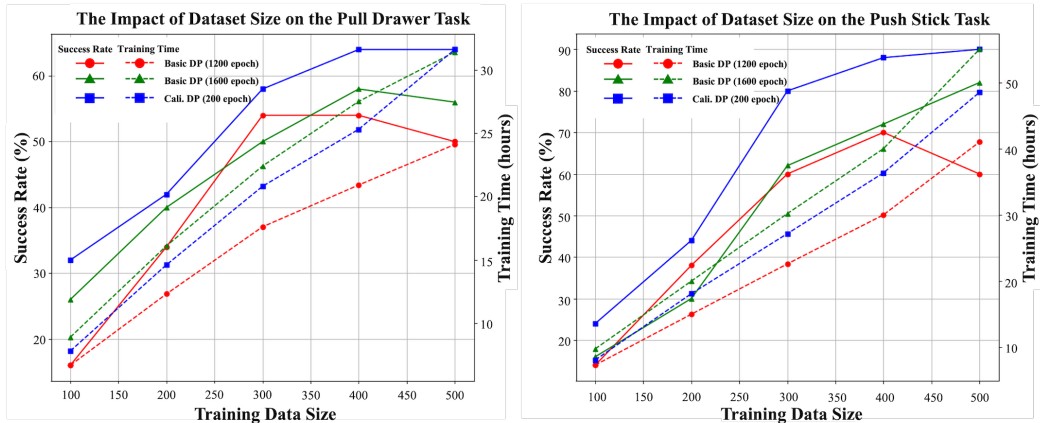

Figure 6: Illustration of the impact of dataset size on two tasks. The dual-vertical-axis diagrams show the trends of the task success rate (left vertical axis) and the time consumed for training (right vertical axis) with the growth of training data size.

enhancing the overall performance. However, the standard Calibration DP enables the calibration network to focus more on learning calibration parameters during the second stage. As a result, it demonstrates superior performance in diverse manipulation tasks from free-view observation.

**View Ablation.** Furthermore, we studied the impact of the observation space of different ranges on the manipulation success rate. In Table 4, we select tasks **P. C.** and **C. F.** which are observed from the front. For convenience, we fix the ranges of the camera in the x-direction and z-direction, and only change the range of y-direction. We randomly select 300 viewpoints within each observation space, collect manipulation actions for training, and conduct tests in the corresponding observation spaces. Judging from the results, the larger the observational space is, the more data samples are likely to be required to learn the actions from different viewpoints. More explorations on view are shown in A.3.

**Sample Ablation.** Finally, we explore the influence of various sampling methods. Specifically, we not only verify two step-wise sampling schedulers (DDPM Ho et al. (2020) and DDIM Song et al. (2021)) applied to the diffusion policy but also extend our Calibration Net to the flow policy, which is a continuous ODE via the Euler sample Lipman et al. (2023). DDPM uses

Table 5: Ablation results for schedulers.

| Method | | P. D. | P. S. | A. B. |
|---|---|---|---|---|
| DP | DDPM | 54% | 60% | 52% |
| | DDIM | 26% | 64% | 46% |
| Cali. DP | DDPM | **58%** | **80%** | **74%** |
| | DDIM | 34% | 68% | 68% |
| FP | Euler | 44% | 54% | 54% |
| Cali. FP | Euler | 50% | 62% | 72% |

probabilistic sampling with random noise introduced in each step to enhance action diversity. DDIM employs deterministic sampling, reusing initial noise and skipping steps to speed up generation. Euler sampling directly maps direct initial noise to target data through learned flows, without explicit denoising. In general, after integrating the Calibration Network, both Diffusion Policy and Flow Policy achieve a considerable increase in the success rate for free-view tasks. DDPM exhibits the highest performance, but demands a lengthy number of sampling steps. It may possess a stronger capability to generate diverse actions with limited training data in the free-view tasks.

## 6 CONCLUSION

This paper introduces a free-view visuomotor policy, enabling the robot to complete manipulation tasks regardless of the observation viewpoints. First, we collect the free-view dataset containing 8 manipulation tasks from different viewpoints in the Isaac Sim environment. Second, we propose a calibration diffusion policy. Based on the basic diffusion policy, it integrates calibration parameters through a calibration network to improve the success rate of free-view manipulation. We conduct extensive of experiments on both the previous methods and the proposed method. These experiments can not only serve as the baseline for the free-view dataset but also prove the effectiveness of the calibration diffusion policy. We believe these free-view manipulation tasks have the potential to enable robots to integrate more seamlessly into society.

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

## A  APPENDIX

### A.1  INTRODUCTION TO DATASET TASKS

Each task is used to test the different abilities of visuomotor policy on robots. The camera's field of view and viewing angle are both defined from the camera's position, pointing toward the robot and the manipulated object; the viewing angle is related to the camera's own position. Here we introduce the composition of the data and the position distribution observed by the camera for each task. The camera position is defined based on the robot's base coordinate system (i.e., the robot's base).

(a) **Pick Cube (P. C.)** (588 episodes): Pick up the cube with a different color from a random position. In this task, the visuomotor policy is required to learn resistance to different colors. Camera random position range: $[1.8\ m < x < 2.5\ m, -1\ m < y < 1\ m, 2\ m < z < 3\ m]$

(b) **Pull Drawer (P. D.)** (752 episodes): Grab the red handle on the front and pull the drawer open. It needs the policy to generate stable linear motion behavior. Camera random position range: $[-2\ m < x < 0.5\ m, -5\ m < y < -3\ m\ |\ 3\ m < y < 5\ m, 1.5\ m < z < 4\ m]$

(c) **Classify Fruits (C. F.)** (984 episodes): Pick the apple, lemon, and plum on the desktop and put them into the corresponding boxes. In this task, the policy is required to possess the capability of accurately classifying diverse fruits. Camera random position range: $[1.5\ m < x < 2.2\ m, -1\ m < y < 1\ m, 1.5\ m < z < 2.0\ m]$

(d) **Push Stick (P. S.)** (675 episodes): Push the blue stick through the passage inside the red walls without touching the walls. This action serves as a means to assess the policy's dexterity in tool manipulation. Camera random position range: $[0.5\ m < x < 2\ m, -2\ m < y < 1\ m, 2\ m < z < 4\ m]$

(e) **Arrange Bottle (A. B.)** (526 episodes): Grab the bottle from the bottom row and put it in a random empty space in the top row. This task demands that policy possess specific observational and space reasoning capabilities. Camera random position range: $[-1\ m < x < -0.3\ m\ |\ 0.3\ m < x < 1\ m, -1\ m < y < -0.5\ m, 1.5\ m < z < 2\ m]$

(f) **Plan Path (P. P.)** (544 episodes): Plan a trajectory to cross the interstices between obstacles and move from the left to the right. This maneuver is specifically designed to rigorously assess the policy's proficiency in obstacle avoidance. Camera random position range: $[1\ m < x < 2.5\ m, -2\ m < y < 0.5\ m, 1.8\ m < z < 3\ m]$

(g) **Strike Ball (S. B.)** (532 episodes): Strike the red ball with the End-Effector, causing the ball to roll and fall into the hole. This task requires the policy to have the ability to predict future states based on the current action. Camera random position range: $[1.6\ m < x < 2.2\ m, -1.5\ m < y < 1.5\ m, 1.5\ m < z < 2\ m]$

(h) **Insert Stick (I. S.)** (584 episodes): Pick up the stick and insert it into the notch at an appropriate orientation. This maneuver rigorously assesses actions' precision in position and orientation generated by the policy. Camera random position range: $[0.5\ m < x < 2\ m, -2\ m < y < 1\ m, 2\ m < z < 4\ m]$

(i) **Real Pick Cup (R. P. C.)** (501 episodes): In the real-world, pick up the paper cup and place it on the white plate. This task examines the effect of free-view manipulation in a real environment. Camera random position range: **About** $[0.7\ m < x < 1.3\ m, -0.4\ m < y < 0.4\ m, 0.2\ m < z < 1\ m]$.

(j) **Real Close Box (R. S. C.)** (502 episodes): In the real-world, move the end-effector to the rear of the box cover and slowly move from left to right to close the box. It examines the ability of the robot to plan continuous movement in the real world. Camera random position range: **About** $[0.7\ m < x < 1.3\ m, -0.4\ m < y < 0.4\ m, 0.2\ m < z < 1\ m]$.

### A.2  VISUALIZATION RESULTS OF COMPARISON

In the visualization results of Figure 7, we observe an interesting phenomenon: while the motion trajectories of the calibration DP and the baseline DP exhibit similar trends, the calibration DP demonstrates superior precision in fine-grained manipulation tasks.

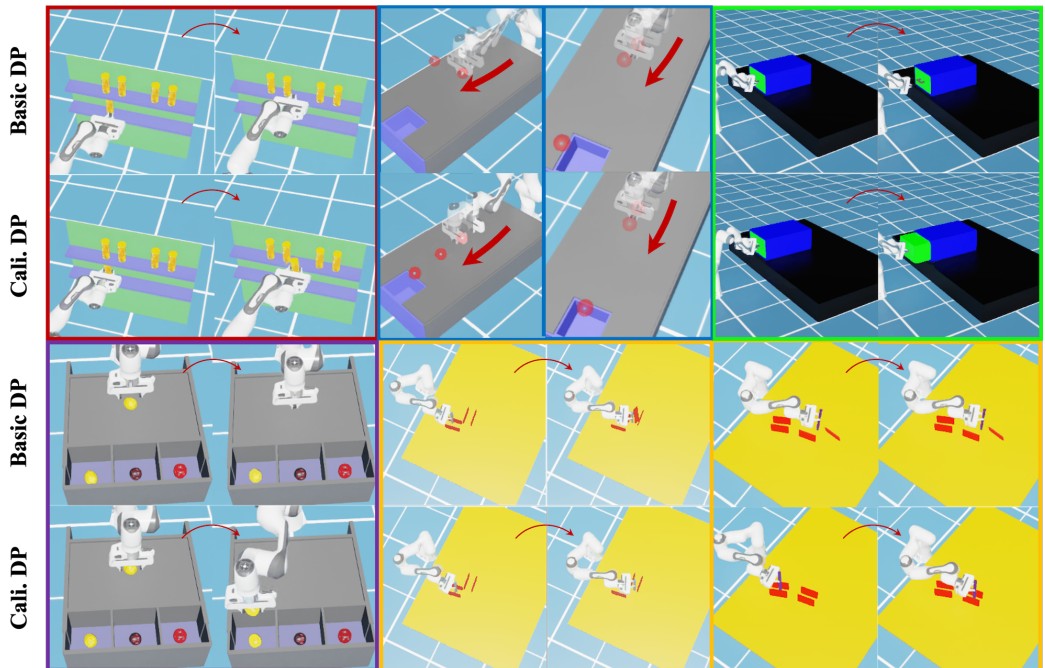

Figure 7: Visualization comparison results. We compare the manipulation actions generated by the basic diffusion policy (DP) and the calibration diffusion policy (Cali. DP) under diverse tasks and viewpoints. The Cali. DP is more refined and has a higher success rate.

## A.3 MORE EXPERIMENTS

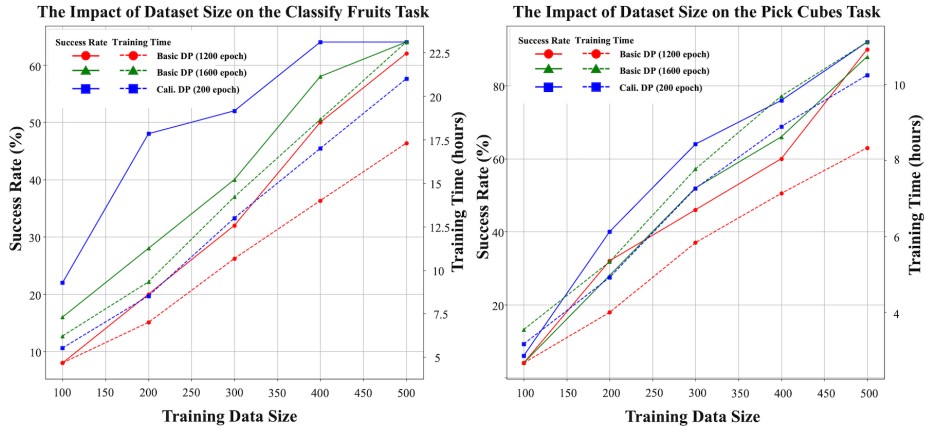

Figure 8: Illustration of the impact of dataset size on two additional tasks. The dual-vertical-axis diagrams show the trends of the task success rate (left vertical axis) and the time consumed for training (right vertical axis) with the growth of training data size.

**Further Data Ablation.** In order to further explore the impact of the size of the dataset on the experimental results, Besides Figure 6, in Figure 8, we show the trend of the task success rate with the size of the dataset under the task **C. F.** and **P. C.**. We obtain similar results. As the dataset grows, the task success rate is higher. The calibration diffusion policy has more advantages when the amount of data is relatively small.

**Further Viewpoints Ablation.** We have discussed in Table 4 the impact of different observation spaces on the free-view task. Specifically, with the same amount of dataset, the smaller the observation space, the higher the manipulation success rate. We further explore the situation where the observation

Table 6: Correspondence results of different training and test space on diffusion policy.

| P. C. Task | Test Space | | |
|---|---|---|---|
| Train Space | $|y| \leq 0.5$ | $|y| \leq 1$ | $|y| \leq 2.0$ |
| $|y| \leq 0.5$ | **70%** | 54% | 42% |
| $|y| \leq 1.0$ | **48%** | 46% | 34% |
| $|y| \leq 2.0$ | 28% | **30%** | 24% |

Table 7: Correspondence results of different training and test space on calibration DP.

| P. C. Task | Test Space | | |
|---|---|---|---|
| Train Space | $|y| \leq 0.5$ | $|y| \leq 1$ | $|y| \leq 2.0$ |
| $|y| \leq 0.5$ | **86%** | 70% | 50% |
| $|y| \leq 1.0$ | **68%** | 64% | 44% |
| $|y| \leq 2.0$ | 30% | 26% | **32%** |

space of training samples differs from that of test samples, as shown in Table 6 and Table 7. We find an interesting point. The model obtained from training samples with a smaller observation space performs better than the model from training samples with a larger observation space, when these two models are tested within a larger observation space. For the P.C. task, for instance, training space $|y| \leq 0.5$ in testing space $|y| \leq 2.0$ has 42% success rate, while training space $|y| \leq 2.0$ in testing space $|y| \leq 2.0$ only has 24% success rate. This is because samples with a smaller observation space enable the model to converge quickly, whereas samples with a larger observation space may fail to converge, leading to inaccurate results. During testing, a model trained with small observation ranges can at least fulfill free-view manipulation tasks within a small observation space.

**Learning Rate Ablation.** In addition, in experiments, we have found that different tasks have varying degrees of sensitivity to the learning rate. We have statistically analyzed the basic diffusion policy trained with different learning rates for three types of tasks, and the test results are shown in Table 8. In the **P.C.** task, cubes of different colors need to be grasped, and in the **C.F.** task, three types of fruits need to be classified. Using a relatively small learning rate of 1e-4 is unable to adapt to the operation of objects

Table 8: Results to show the impact of learning rate on diffusion policy.

| lr | P. C. | C. F. | P. S. |
|---|---|---|---|
| 1e-4 | 12% | 20% | **60%** |
| 5e-4 | 46% | **32%** | 40% |
| 8e-4 | **52%** | 30% | 24% |

with different surface features (colors). But in the **P.S.** task, only the position of the red obstacle is changed without changing in color. It is necessary to accurately control the movement trajectory of the end effector. In this task, using a large learning rate makes it impossible to achieve accurate control.

**Calibration Noise Ablation.** In the simulation environment, we directly obtain accurate calibration parameters (from the camera coordinate system to the robot coordinate system) and use them as part of the input to enhance the policy's robustness against viewpoint variations. However, it is undeniable that we may not be able to acquire accurate calibration parameters in the real world. Therefore, here we further explore the impact of calibration errors on our Calibration DP by adding noise to the calibration parameters. During training, we only introduced Gaussian noise of different scales (no noise ; $\mu = 0$, $\sigma = 0.01$; $\mu = 0$, $\sigma = 0.1$) in the second stage of training. During testing, we evaluated the model's performance under different calibration errors. The specific results are shown in Tables 9, 10, and 11.

Table 9: Experimental Results of "No Training Noise".

| Calibration DP | Push Stick | Classify Fruits | Arrange Bottle |
|---|---|---|---|
| $\mu = 0, \sigma = 0$ (base) | 80% | 52% | 74% |
| $\mu = 0, \sigma = 0.05$ | 66% | 24% | 48% |
| $\mu = 0, \sigma = 0.1$ | 52% | 2% | 32% |
| $\mu = 0, \sigma = 0.2$ | 16% | 0% | 0% |
| $\mu = 0, \sigma = 0.5$ | 0% | 0% | 0% |

We can observe that if no noise is added during the training process, the noise in the testing process will have a significant impact on the results. If a large noise $\mu = 0$, $\sigma = 0.1$ is added during the training process, the impact of noise on the results will be reduced, but the performance improvement brought by Calibration DP will also diminish. It seems that the calibration parameters have become ineffective, and the observed improvement is instead caused by the increased model complexity. Therefore, our Calibration DP has relatively strict demands for the accuracy of calibration parameters.

Table 10: Experimental Results of "Training Noise ($\mu = 0, \sigma = 0.01$)".

| Calibration DP | Push Stick | Classify Fruits | Arrange Bottle |
|---|---|---|---|
| $\mu = 0, \sigma = 0$ (base) | 78% | 54% | 66% |
| $\mu = 0, \sigma = 0.05$ | 78% | 48% | 66% |
| $\mu = 0, \sigma = 0.1$ | 66% | 40% | 64% |
| $\mu = 0, \sigma = 0.2$ | 60% | 28% | 58% |
| $\mu = 0, \sigma = 0.5$ | 16% | 0% | 30% |

Table 11: Experimental Results of "Training Noise ($\mu = 0, \sigma = 0.1$)".

| Calibration DP | Push Stick | Classify Fruits | Arrange Bottle |
|---|---|---|---|
| $\mu = 0, \sigma = 0$ (base) | 72% | 44% | 64% |
| $\mu = 0, \sigma = 0.1$ | 75% | 44% | 62% |
| $\mu = 0, \sigma = 0.2$ | 72% | 40% | 60% |
| $\mu = 0, \sigma = 0.5$ | 52% | 18% | 32% |

Table 12: Multi-view stable-object results on simulation tasks (300 episodes).

| Method | P. C. | P. D. | C. F. | P. S. | A. B. | P. P. | S. B. | I. S. |
|---|---|---|---|---|---|---|---|---|
| BC-T Florence et al. (2022) | 0% | 8% | 28% | 44% | 16% | 0% | 6% | 0% |
| DP Chi et al. (2023) | 52% | 80% | 46% | 76% | 62% | 22% | 32% | **16%** |
| DP3 Ze et al. (2024b) | 62% | 36% | **68%** | 74% | 48% | 4% | **48%** | 0% |
| **Calibration DP** | **68%** | **74%** | 66% | **92%** | **80%** | **34%** | 36% | **16%** |

**Fixed Multi-View Evaluation.** Based on the results of Table 2, in the simulation environment, we also define a multi-view stable-object evaluation situation. On one hand, the observation viewpoints are fixed on multiple positions, and one of these viewpoints is selected for observation in each episode. On the other hand, the manipulated object is manually placed by ours so that it appears as close as possible to the center of the field of view and is not occluded. This situation is used to avoid individual cases where the object itself is difficult to observe or is at the edge of the observation point in the free-view free-object situation. The training data also uses 300 episodes from the free-view viewpoints. The multi-view stable-object results are shown in Table 12. Through the selection of viewpoints and object positions during the testing phase, the task success rate can be improved. This situation might better align with the practical application contexts of robots. After all, when executing manipulation tasks, robots typically choose a vantage point from which they can clearly observe the target object.

## A.4 REAL-WORLD EXPERIMENTS

As for real-world experiments, we similarly introduced variations in camera viewing angles. In read-world, changes in viewing angles also lead to variations in complex backgrounds, and we found that background removal exerts a significant impact on the results. As shown in Table 13, by employing methods such as depth map and color filtering, as well as semantic segmentation, the experimental results are substantially improved after background removal. We attribute this phenomenon to the inability of the image feature extraction network to extract effective foreground information from complex backgrounds, rather than the failure of the calibration network. 5299164

Table 13: The Impact of Background Removal in Real-World.

| Task | DP | Calibration DP |
|---|---|---|
| R.P.C with Background | 10% | 18% |
| R.P.C without Backgroud | 38% | 46% |
| R.C.B with Background | 6% | 6 % |
| R.C.B without Backgroud | 52% | 58 % |

## A.5 COMPARISON WITH THE MULTI-VIEW METHOD

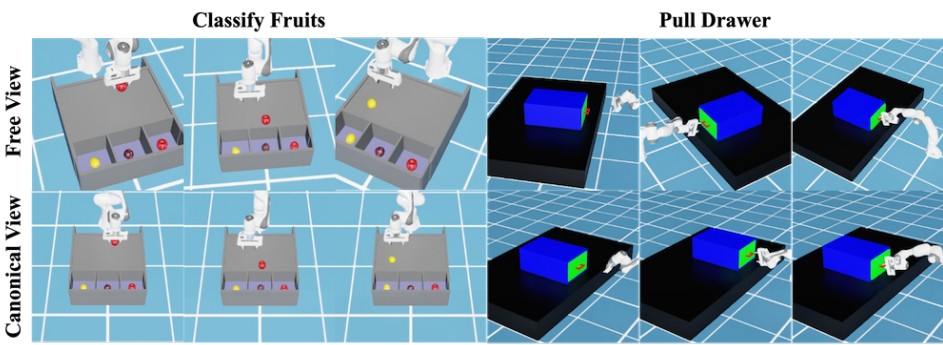

Figure 9: Visualization of Free View and Canonical View for generating multi-view data.

Some Multi-View methods such as Maniwhere Yuan et al. (2024) and ReViWo Pang et al. (2025) , implicitly align the image features from multiple viewpoints with those of the canonical view. We regenerated image pairs of Free View and Canonical View in the ISAAC-Sim simulation environment as shown in Fig. 9, and attempted to reproduce Maniwhere Yuan et al. (2024). Maniwhere is a reinforcement learning method that requires constructing a reinforcement learning environment, which differs from our approach of leveraging a diffusion policy for imitation learning. We only adopted the STN Module in Maniwhere to align the features of Free View with those of the Canonical View, and then we used the BC-T Transformer module to generate actions (STN + BC-T) and the Diffusion Policy action generation policy (STN + DP) separately.

Table 14: Comparison with the Multi-View Method Results.

| Method | C. F. | P. D. |
|---|---|---|
| BC-T | 18% | 0% |
| Maniwhere STN + BC-T | 22% | 0% |
| DP | 32% | 54% |
| Maniwhere STN + DP | 44% | 50% |
| Calibration DP | 52% | 58% |

As shown in Table. 14, we selected the Classify Fruits and Pull Drawer tasks for experiments. From the results, the Maniwhere STN improves the success rate of free-viewpoint manipulation tasks, but under our experimental setup, it does not outperform Calibration DP. STN is an implicit feature alignment method that requires additional Canonical View information, whereas Calibration DP requires additional calibration information. In real-world applications, calibration information may be subject to noise and errors, whereas Canonical View images do not face such issues. However, during the data collection process, it is necessary to ensure that Canonical View does not change.

## A.6 LIMITATION AND FUTURE WORK

We propose that the calibration diffusion strategy relies on the calibration parameters between the robot and the camera. Although we demonstrated that the calibration diffusion policy remains effective in real-world tasks, obtaining accurate calibration parameters is still a rather cumbersome step in practical tasks. In the future, we will attempt to weaken the calibration parameters, enabling the visuomotor policy to complete manipulation tasks through implicit learning from different perspectives.

