# OpenReview forum: "Free-View Robot Manipulation: Visuomotor Policy by Calibration Diffusion"
_ICLR.cc/2026/Conference — Submitted to ICLR 2026_

### Official Review · Reviewer_dmaV · 2025-10-25

**Soundness:** 2
**Presentation:** 2
**Contribution:** 2
**Rating:** 2
**Confidence:** 4

**Summary:**

This paper introduces the free-view robot manipulation task to overcome the fixed viewpoint limitation in current visuomotor policies. The authors construct a new dataset with 8 tasks and over 5,000 simulation episodes, each from varying viewpoints. Their solution is a calibration diffusion policy, which uses a novel calibration network and a two-stage training curriculum to achieve effective, viewpoint-invariant manipulation. The method's success is validated through extensive experiments, establishing a strong new benchmark for this problem.

**Strengths:**

1. the topic the paper focuses on is an interesting and crucial problem in current visuomotor policy for robot manipulation.
2. the paper is well-written with little typos.
3. the author try to build a benchmark to systematically analyze the free-view robot manipulation problem.

**Weaknesses:**

1. The contribution of the proposed dataset is undermined by its limited scale and lack of detailed specification. The paper would be strengthened by providing a comprehensive description of the camera pose distribution used for data collection, including the ranges for azimuth, elevation, and distance from which viewpoints were sampled.
2. The method lacks of specific design for the free-view robot manipulation problem. The author simply add the calibration parameters to the current diffusion policy pipeline without any insight of this problem.
3. The statement of the two-stage training pipeline is confused.  The purpose of introducing random noise states in the first stage is unclear and requires elaboration. Please clarify the  training strategy with figure 3.
4. The experimental evaluation lacks critical details regarding the viewpoint split between training and testing. It is essential to clarify whether the tested viewpoints were entirely unseen during training or were simply a held-out set from the same distribution. Please present more details of experiment evaluation.
5. The most significant weaknesses of the paper is lack of real-world experiments. The absence of real-world experiments leaves the method's practicality, robustness to perceptual noise unproven.

**Questions:**

Please see the weaknesses.

**Details Of Ethics Concerns:**

No ethic concerns.

---

> ### Author Response · Authors · 2025-11-17
>
> Thanks for your comments
>
> **W1**: Limited scale and no detailed specifications reduce the dataset’s value.
>
> **A**: We have updated a PDF version and added two real-world tasks to the dataset.
> The dataset now includes 8 simulation tasks along with 2 real-world tasks.
> For the simulation tasks, the data has been generated by using scripts, and users are welcome to create additional data through these scripts as well. In APPENDIX A.1, we have included information regarding the camera pose distribution for each task.
> For real-world tasks, camera calibration is required for each episode. We provide two tasks, each with 500 episodes, all of which contain recalibrated camera parameters.
>
> **W2**: Lacks of specific design & **W3**: Random Noise States in Fig3 is unclear.
>
> **A**: It seems that there might be a slight misunderstanding of the diffusion-based methods.
> To clarify, diffusion models [1] inherently generate images by gradually denoising random noise images through Denoising Diffusion Probabilistic Models (DDPM). The variability present in the random noise contributes to enhanced generalization of the generated images.
>
> Regarding the diffusion policy approach [2], this is part of our training process in the first stage. It generates generalizable actions by progressively denoising random noise (as illustrated by the random noise states shown in Fig. 3) using DDPM.
>
> Importantly, in this work, we do not merely extract features from calibration parameters for integration into the neural network model. Instead, inspired by ControlNet [3], we utilize a Calibration Net to guide the denoising direction, thereby regulating action generation to better adapt to free-view scenarios.
>
> Thank you for your understanding. If you have any questions, please feel free to communicate with us at any time.
>
> [1] Denoising diffusion probabilistic models, NIPS, 2020.
>
> [2] Diffusion policy: Visuomotor policy learning via action diffusion, IJRR, 2023.
>
> [3] Adding conditional control to text-to-image diffusion models, ICCV, 2023.
>
>
> **W4**: Distribution of viewpoints for training and testing.
>
> **A**: Thank you for your question. This is indeed something we need to clarify.
> The test viewpoints remain entirely unseen during the training phase.
> Our training set is generated randomly within a specified range of camera pose angles, and similarly, the test set is also re-randomized within the same range of camera poses.
> It is nearly impossible to produce two identical camera positions.
> Consequently, the camera viewpoint for each episode in the training set can be regarded as unique when compared to that of each episode in the test set.
>
> **W5**: About Real-World Experiment.
>
> **A**: This problem is also a concern of ours. We are pleased to inform you that we have uploaded a new version, which now includes two additional real-world tasks. The results indicate that our method remains effective in practical experiments.
>
> In APPENDIX A.3, we have conducted Calibration Noise Ablation experiments as well: on one hand, Gaussian noise was introduced to the calibration parameters of the training set; on the other hand, Gaussian noise was applied to the calibration parameters during the testing process.
> The experimental findings suggest that our model exhibits a degree of robustness against calibration noise.

---

### Official Review · Reviewer_3urr · 2025-10-28

**Soundness:** 2
**Presentation:** 2
**Contribution:** 2
**Rating:** 4
**Confidence:** 5

**Summary:**

This work aims to deal with the viewpoint generalization problem for robotic manipulation. The author constructs a free-view dataset, which encompasses 8 tasks with over 5,000 episodes sourced from the Isaac Sim simulation environment. Then, a calibration diffusion policy training method is introduced to enhance the adaptability of the diffusion policy to different viewpoints. To verify their method, the author does some experiments based on Issac Gym.

**Strengths:**

1. The method is novel and the problem you concern is important.
2. For simulation, the author introduce well-designed free-view settings.

**Weaknesses:**

1. No multi-view related method is included as baseline. There have been many methods proposed to deal with viewpoint disturbance setting, like RoboUniview[1], Maniwhere[2], ReViWo[3] or MV-MWM[4]. However, no kind of such baselines is included.

2. We desire a real-world experiment to validate the effectiveness of your method.

3. For real-world tasks, despite using self-collected data, we wonder if some open-source multi-view data could also be used to better improve your model capability.

References:
[1] RoboUniView: Visual-Language Model with Unified View Representation for Robotic Manipulation.
[2] Learning to Manipulate Anywhere: A Visual Generalizable Framework For Reinforcement Learning.
[3] Learning View-invariant World Models for Visual Robotic Manipulation.
[4] Multi-View Masked World Models for Visual Robotic Manipulation.

**Questions:**

See weaknesses.

---

> ### Author Response · Authors · 2025-11-17
>
> Thanks for your comments.
>
> **W1**: Compare with the multi-view related method.
>
> **A**: MV-MWM and RoboUniview are indeed multi-view methods, which require the simultaneous input of multiple images. In contrast, our approach is designed to work effectively with just a single view for both training and testing.
> Maniwhere and ReViWo's innovative policy introduces an additional Canonical View during the training process. This approach enables image features from various viewpoints to align more closely with those of the Canonical View.
> During testing, only a single view is required.
>
> We generated image pairs of Free View and Canonical View within the ISAAC-Sim simulation environment, and we made an effort to reproduce the work done in Maniwhere.
> Maniwhere is a reinforcement learning method that necessitates the construction of a specific reinforcement learning environment, which contrasts with our approach that utilizes a diffusion policy for imitation learning.
> In our study, we chose to adopt only the STN Module from Maniwhere to align the features of Free View with those of Canonical View. Subsequently, we employed other modules (Transformer and Diffusion) and policies to generate actions.
> The detailed results can be found in APPENDIX A.5.
>
>
> **W2**: About Real-World Experiment.
>
> **A**: We are pleased to announce that we have uploaded a new version, which now includes two additional real-world tasks.
> Based on the results, it appears that our method is also effective in real-world experiments.
>
>
> **W3**: About using open-source multi-view data.
>
> **A**: Our work is centered around training a small-model policy, which we conduct using a single RTX 4090.
> Typically, the image feature extraction network is tailored to address a specific task.
> While open-source multi-view data can be utilized for pre-training  (just like pre-training on ImageNet), it ultimately requires that training occurs within the confines of a single task.
>
> We believe that when employing a multimodal large model, harnessing open-source multi-view data could prove to be more effective, and this larger model may also exhibit some resilience against view interference.

---

### Official Review · Reviewer_5TS4 · 2025-10-30

**Soundness:** 3
**Presentation:** 2
**Contribution:** 2
**Rating:** 4
**Confidence:** 4

**Summary:**

This paper introduces a Free-View Robot Manipulation framework that enables robots to perform visuomotor manipulation tasks from arbitrary camera viewpoints. It builds a new Free-View Dataset (8 tasks, 5000+ episodes) within Isaac Sim and proposes a Calibration Diffusion Policy (Cali. DP) that integrates camera calibration parameters into diffusion-based visuomotor policy learning. Extensive simulation experiments demonstrate improved generalization to viewpoint variations compared to baselines like ACT, Diffusion Policy, and Flow Policy.

**Strengths:**

1. The paper addresses an important and underexplored challenge (viewpoint generalization) in visuomotor policy learning, with clear motivation and relevance to real-world deployment.
2. It constructs a well-designed free-view dataset with diverse manipulation tasks and calibration annotations, providing a valuable benchmark for future research.
3. Experimental results are comprehensive, including comparisons with strong baselines, ablation studies, and multiple evaluation settings, demonstrating meaningful performance gains.

**Weaknesses:**

1. All experiments are conducted purely in simulation (Isaac Sim) without any real-world validation; thus, the claimed generalization to varying viewpoints remains unverified under physical-world noise and imperfections.
2. The proposed Calibration Diffusion Policy shows limited methodological novelty. It mainly adapts ControlNet-like conditioning to diffusion policy rather than introducing a fundamentally new idea.
3. The dataset’s task diversity and complexity are relatively low (no deformable or dual-arm tasks), limiting its generalizability and practical relevance.
4. The paper’s claim that existing methods fail under viewpoint changes is insufficiently supported; it lacks direct comparisons with modern multi-view or hybrid-view frameworks (e.g., OpenVLA, RDT, or $\pi_0$) that already handle varying viewpoints effectively.

**Questions:**

1. Can the proposed Calibration DP be validated on a real robot to assess robustness under real-world calibration noise?
2. How does the method perform when calibration parameters are inaccurate or unavailable, as often occurs in practice?
3. Could comparisons be extended to recent Vision-Language-Action systems or hybrid-view approaches to strengthen the motivation?
4. Is the calibration network truly necessary, or could similar improvements be achieved with modern feature alignment or camera-pose estimation modules?

---

> ### Author Response · Authors · 2025-11-17
>
> Thanks for your comments.
>
> **W1 & Q1**: About real-world tasks.
>
> **A**: We are pleased to announce that we have uploaded a new version, which now includes two additional real-world tasks. In real environments, each episode requires camera calibration to obtain the necessary parameters and build datasets.
> As shown in Table 2 and Figure 5, we are happy to share that our method continues to be effective in these real-world applications.
>
>
> **W2**：Limited methodological novelty.
>
> **A**：We’re truly inspired by ControlNet and grateful for the foundation it provides. Building on that, we’ve had the opportunity to introduce a few meaningful enhancements of our own. First, we’ve integrated camera-robot calibration as a core conditional input for diffusion-based robot manipulation—an approach that, to the best of our knowledge, is being explored in this field for the first time. Second, while there are some similarities, our design differs in important ways: we’ve adapted the data scale and training strategy with the practical needs of robotic systems in mind, aiming to make the method more effective. Third, we’ve created the novel free-view dataset and benchmark, and we’re excited to share them with the community, hoping they can help support and inspire future research.
>
>
> **W3**: The dataset is not complex enough.
>
> **A**:  In the new version, we’re excited to introduce real-world data, making the Free-View dataset now include 10 diverse tasks.
>
> In the simulation data, we’ve designed eight different tasks where, in each episode, the position of the object being manipulated varies, and the camera perspective shifts as well. Some tasks even involve different types of objects—such as “pick cube” and “classify fruits”—offering rich variability for training and evaluation.
>
> For the real-world data, we’ve put in significant effort to collect data from two real-world robotic tasks. In every episode, the camera position is adjusted, and an independent camera calibration is carefully performed. We believe these data can meaningfully support research in free-view robot manipulation.
>
> Additionally, we’ve noticed that most existing robot manipulation datasets are collected using only a single robot configuration. It’s rare to find datasets that include data across different robot setups, and we hope ours can help fill that gap and inspire broader exploration.
>
>
> **W4 & Q3**: Could compare to VLA (Vision-Language-Action) system.
>
> **A**: We sincerely apologize for our inability to compare our model with VLA systems.
> On one hand, our approach emphasizes the adaptability of small visuomotor policies to various camera viewpoints, ensuring that all comparisons are conducted fairly among small-model baselines.
> In terms of training resources, we currently do not have access to GPU clusters and have only been able to perform training on a single NVIDIA RTX 4090 GPU.
>
> On the other hand, VLA systems focus on task generality by utilizing internet-scale training data and a significantly larger number of model parameters. In contrast, our model is designed with considerably fewer parameters.
>
>
> **Q2**: About inaccurate calibration parameters.
>
> **A**: For our simulation experiments, as detailed in APPENDIX A.3, we have conducted Calibration Noise Ablation experiments. On the one hand, we introduced Gaussian noise to the calibration parameters of the training set. On the other hand, Gaussian noise was also added to the calibration parameters during the testing process.
> Our findings indicate that small-scale noise has minimal impact on our method.
>
> For real-world experiments, it is important to acknowledge that calibration parameters in the real world may inevitably include some degree of calibration noise.
> Nevertheless, our method continues to demonstrate a commendable level of robustness in practical applications.
>
>
> **Q4**: The necessity of calibrating the network.
>
> **A**: We thoughtfully incorporate calibration parameters and utilize a calibration network, which plays a crucial role in our design.
> While we believe that other methods, such as feature alignment, can also be effective in adapting to viewpoint changes, these approaches may necessitate additional information and more complex network structures.
> For example, Maniwhere (we replicated it in APPENDIX A.5) requires an additional Canonical View.

---

### Official Review · Reviewer_K2pn · 2025-11-01

**Soundness:** 3
**Presentation:** 3
**Contribution:** 3
**Rating:** 6
**Confidence:** 3

**Summary:**

This paper proposes a new framework for viewpoint-invariant robot manipulation by introducing a calibration diffusion policy. The authors also propose a free-View robot manipulation dataset constructed in the Isaac Sim environment, covering eight manipulation tasks. The method integrates a calibration network into the diffusion policy to adapt to variable camera viewpoints. Extensive experiments and ablation studies are conducted to evaluate the approach against several baselines, including BC-T, ACT, Diffusion Policy, Flow Policy, and DP3.

**Strengths:**

1. The paper identifies a relevant and underexplored issue, that is the sensitivity of visuomotor policies to camera viewpoints, and formalizes the “free-view manipulation” as a benchmark task. This is novel to me.
2. The Free-View dataset is clearly structured, includes calibration parameters for each episode, and provides an important testbed for evaluating viewpoint generalization.
3. The experiments cover comparisons with multiple strong baselines, data size ablations, structure ablations, and calibration noise analyses. The evaluation protocol is clearly explained. and overall it is generally well organized and includes clear visualizations.

**Weaknesses:**

1. The entire evaluation is done in simulation of Isaac Sim. There lack of physical robot validation limits the practical credibility of the results.
2. The description of the calibration network could be improved to make it more clear. The method section is difficult to follow, with unclear notations and redundant equations. Key architectural details are missing
3. The method requires accurate calibration parameters between the robot and each camera. As the authors themselves admit in Section A.3, this dependency makes the approach impractical for real-world scenarios, where calibration is noisy or unavailable.

**Questions:**

1. How is the camera calibration represented in real-world tasks, and could the model generalize to approximate or partially incorrect calibrations?
2. How scalable is the proposed two-stage training strategy when the number of viewpoints or tasks increases significantly?
3. Could the calibration features be learned implicitly, rather than explicitly provided?

---

> ### Author Response · Authors · 2025-11-17
>
> Thanks for your comments.
>
> **W1 & Q1**: About real-world tasks and incorrect calibrations.
>
> **A**: We have uploaded a new version, which includes two additional real-world tasks.
> Camera calibration parameters are represented by the camera’s translation and quaternion in the robot coordinate system.
>
> Additionally, we place significant importance on addressing the issue of incorrect calibrations. In APPENDIX A.3, we conducted Calibration Noise Ablation experiments. On one hand, Gaussian noise was introduced to the calibration parameters of the training set; on the other hand, Gaussian noise was also added to the calibration parameters during testing. The experimental results indicate that our model demonstrates a commendable level of robustness against calibration noise.
>
> **W2**: A more detailed description of the "method section" is required.
>
> **A**: Thank you for your patient review. We have submitted the updated version and carefully refined the "Method" section, including clear explanations for each notation. We hope these improvements make the content easier to follow. If you have any further suggestions or feel anything could be clarified further, we would be happy to hear from you.
>
> **W3**: Calibration parameters are difficult to obtain in the real world.
>
> **A**: We have been working on this project, investing a great deal of effort to collect datasets for two tasks in the real world. Each episode underwent an independent camera calibration, and we completed the experiments in the real world. However, the noise in the calibration parameters of the real world is still inevitable. But from the results, our method shows a certain degree of robustness to noise.
>
> In addition, obtaining the calibration parameters of free-view cameras in the real world is extremely complex. We believe that the real-world data we collected with calibration parameters will provide effective assistance for the research of free-view manipulation.
>
>
> **Q2**:  About the scalability of the two-stage training strategy.
>
> **A**: In our dataset, each episode corresponds to a specific viewpoint. We believe that increasing the number of viewpoints will enhance the volume of data available for model training. For the same task, we think that adding more viewpoints is akin to providing additional data.
> However, for different tasks, it may be necessary to retrain the policy.
> This is because our method emphasizes adapting to camera viewpoints rather than generalizing across various tasks.
>
>
>
> **Q3**: Could the calibration features be learned implicitly?
>
> **A**: Some multi-view methods do indeed align image features through implicit techniques; however, they may require additional network structures and information. For instance, Maniwhere (which we have replicated in APPENDIX A.5) necessitates an extra Canonical View.
> We are also actively engaged in follow-up work related to this topic.

---

### Author Response · Authors · 2025-11-17
**Upload a new version of the PDF**

Thank you for the valuable comments from the reviewers. We sincerely appreciate their concerns regarding real-world experiments, and we want to assure you that we have been actively working on applying the Calibration Diffusion Policy method to real-world tasks.

In response to these important points, we have updated the PDF to a new version. The key differences from the previous version are as follows:
1. Two real-world tasks have been introduced: Real Pick Cup and Real Close Box, as illustrated in Figure 2. For these tasks, we conduct camera calibration prior to each data collection session and gather a total of 500 trajectories for each task. The corresponding experimental results are presented in Table 2 and Figure 5.
2. We have relocated the description of the tasks within the dataset to APPENDIX A.1 and included the range of camera positions applicable for free-view settings.
3. In Appendix A.5, we generate image pairs for both Free-View and Canonical-View scenarios, along with a comparative analysis against multi-view methods.
4. We have refined Method Section to make it clearer and more understandable.

---

### Meta-Review · Area_Chair_rzN1 · 2026-01-06

**Summary:**

All four reviewers recognized the importance of viewpoint generalization in visuomotor policies and appreciated the free-view dataset contribution. However, they raised substantial concerns supporting rejection. The primary issues were: complete absence of real-world validation in the original submission, limited methodological novelty with the calibration network appearing as a straightforward ControlNet adaptation without significant new insights, and concerns about dataset scale and missing experimental details like camera pose distributions. Reviewer dmaV gave the lowest rating (2), citing lack of specific design insights and unclear training strategies. Reviewers K2pn, 5TS4, and 3urr gave moderate ratings (6, 4, 4) but shared concerns about simulation-only validation and missing comparisons with modern multi-view methods.

Despite efforts addressing reviewer concerns, the core methodological contribution remains limited. Essentially an adaptation of existing techniques without substantial innovation specific to free-view manipulation. While the dataset has merit, it suffers from limited scale and complexity. Multi-view method comparisons remain incomplete with only partial Maniwhere replication rather than comprehensive evaluation against recent approaches. The real-world experiments, though valuable, are too limited to fully validate practical effectiveness. Even with rebuttal improvements, the combination of limited novelty, incomplete validation, and missing state-of-the-art comparisons would have unlikely shifted the reviwers opinion during the rebuttal.

**Reviewer Concerns:**

The authors made substantial efforts by adding two real-world tasks with 500 trajectories each, calibration noise ablation studies, camera pose distributions, and a Maniwhere replication. These additions address several concerns, particularly real-world validation and robustness to noise. However, limitations remain: the real-world experiments are limited compared to eight simulation tasks, methodological novelty concerns persist as authors confirmed they adapted ControlNet without fundamental algorithmic innovations, and comparisons with VLA systems were not feasible due to resource constraints. The clarification that test viewpoints were unseen but sampled from the same distribution somewhat limits generalization claims.

**Reviewer Scores:**

Reviewer K2pn (6) would likely maintain or slightly improve to 6-7 given real-world validation. Reviewer 5TS4 (4) might increase to 5-6 as their primary concern was addressed. Reviewer 3urr (4) would likely reach 5 but remain below threshold due to incomplete multi-view comparisons. Reviewer dmaV (2) might increase to 3-4 but remain negative given persistent methodological concerns despite acknowledging limited familiarity with diffusion-based policies.

---

### Decision · Program_Chairs · 2026-01-26

Reject